# Successful Implant Placement via Simultaneous Nasal Floor Augmentation in an Inferior Meatus Pneumatization Case

**DOI:** 10.3390/medicina59020357

**Published:** 2023-02-13

**Authors:** Won-Bae Park, Gazelle Jean Crasto, Wonhee Park, Ji-Young Han, Philip Kang

**Affiliations:** 1Private Practice in Periodontics and Implant Dentistry, Seoul 02771, Republic of Korea; 2Division of Periodontics, Section of Oral, Diagnostic and Rehabilitation Sciences, Columbia University College of Dental Medicine, New York, NY 10032, USA; 3Department of Dentistry, Division of Dentistry, College of Medicine, Hanyang University, Seoul 04763, Republic of Korea; 4Department of Periodontology, Division of Dentistry, College of Medicine, Hanyang University, Seoul 04763, Republic of Korea

**Keywords:** dental implant, inferior meatus pneumatization, nasal cavity, nasal floor elevation

## Abstract

Partially edentulous patients who present with inadequate bone height in the posterior maxillary can predictably be rehabilitated with lateral wall sinus augmentation and subsequent implant placement. However, the sinus augmentation is defined by variations observed in the anatomical presentation of the maxillary sinus. Herein, we describe a case study managing sinus augmentation when a rare anatomic variant termed inferior meatus pneumatization was observed. A 65-year-old female patient presented, wherein the inferior meatus of the nasal cavity was located directly above the maxillary posterior dentition as opposed to the maxillary sinus. The clinically atrophied maxilla was rehabilitated by employing nasal floor elevation, bone augmentation, and simultaneous implant placement. Post-operatively, no sino-nasal complications were recorded. Subsequently, 8 months after the initial procedure, osteointegration of the implants along with the presence of vital bone was observed. The patient posterior occlusion in the upper right quadrant was rehabilitated by engaging the stable implants with a cement-retained fixed final prosthesis. Follow-ups recorded for up to 2 years demonstrated no further complications. The case report demonstrates diagnosis, appropriate treatment, and management of inferior meatus pneumatization and a viable surgical approach for augmentation and implant treatment.

## 1. Introduction

Rehabilitation of the posterior maxillary edentulous areas with implant placement is often plagued with insufficient bone volume caused due to alveolar bone loss and sinus pneumatization [1]. Grafting the floor of the maxillary sinus, commonly known as lateral wall sinus augmentation, is routinely employed to increase the alveolar bone height. The average success rate of the implants placed in the augmented sinus with lateral window specifically was 91.8%, on average [2]. However, the outcomes of this procedure are often influenced by the specific surgical technique and the anatomic variations observed in patients. Inferior meatus pneumatization, one such variant, influences the potential surgical approach for sinus augmentation. The ethmoid air cells present at birth continue to change and develop with age. Enlargement of the posterior cells continues and occasionally creates larger posterior cells than anterior [3]. The condition is defined when the inferior meatus pneumatization inferiorly displaces the nasal cavity in the posterior direction, such that the nasal cavity now occupies the space of the maxillary sinus and is above the maxillary posterior teeth [4]. Typically observed in 3% of patients, this anatomic variant is problematic for sinus augmentation and subsequent implant placement [5]. Two-dimensional radiographic views do not provide enough documentation to detect these changes in the maxillary sinus. Furthermore, due to the rarity of the anatomical finding, clinicians may accidentally penetrate the nasal cavities during surgery. Cone-beam computed tomography (CBCT) provides a thorough evaluation, wherein the variation is correctly identified and the extent of the inferior meatus pneumatization, a rare condition, can be accurately detected and measured. Previous studies, utilizing canines, demonstrated normal mucoperiosteum and bone engaging the implants placed in the nasal cavity. However, the long-term success and survival of those implants were recorded to be 70% at 5 years [6]. Thereby, adequate detection of anatomic variations and appropriate treatment can alleviate the failures observed in implants placed in the nasal cavity.

The purpose of this case report is to describe and detail the nasal floor augmentation procedure to circumvent the inferior meatus pneumatization noted. Furthermore, the goal is to demonstrate that successful rehabilitation of the patient’s right maxillary quadrant can be accomplished with implant placement and restoration without any complications for up to 2 years.

## 2. Case Descriptions

### 2.1. Patient Information

A 65-year-old female patient presented to a private dental office in Seoul, Republic of Korea, with a chief concern, to rehabilitate her posterior occlusion with the aid of dental implants. The patient self-reported as a non-smoker, with no systemic health conditions and no medications. Her medical and dental history were obtained, and the patient presented without any nasal or sinus-related symptoms. A physical head and neck examination was performed. Clinical examination revealed no intra- or extra-oral swelling or asymmetry. Past dental history revealed the loss of posterior upper right dentition due to periodontitis with concomitant vertical and horizontal bone resorption (Figure 1a) depicted in the panoramic radiograph. In comparison to the left dentate maxillary quadrant, the right maxillary bone height is significantly decreased due to vertical bone loss. The radiographic trace of the floor of the right maxillary sinus appears well-defined. However, subsequent cone beam computed tomography (CBCT) (PaX-i3D Smart, Vatech, Seoul, Republic of Korea) imaging depicts a closely positioned nasal cavity located directly apical to the ridge (Figure 1b). The coronal view clearly shows the most coronal portion of the maxillary sinus at the upper-left corner of Figure 1b, while the bucco–palatal dimension of the remaining ridge showed deficient lateral volume. The inferior meatus of the nasal cavity extended to the posterior maxilla and the lateral breadth appeared wide (Figure 1c).

### 2.2. Procedure for Surgical Technique and Post-Operative Management

All procedures were performed in accordance with the ethical rules and the principles of the Declaration of Helsinki. The patient signed an informed consent to participate in the project and to have the case published in an academic journal. 

Lateral window sinus augmentation with simultaneous implant placement, as described by Tatum [7], was planned. Briefly, the procedure involves creating a lateral window on the lateral nasal wall giving access into the inferior meatus pneumatization, followed by intranasal bone augmentation. The surgery, including the extraction of teeth with recurrent caries, was performed under local anesthesia. Access to the buccal maxillary wall was achieved via mucosal mid-crestal incisions, including anterior and posterior vertical incisions. Full thickness mucoperiosteal flaps were reflected buccally and palatally (Figure 2a). An oval-shaped bone window measuring 2.0 mm × 1.5 mm was outlined and prepared using a diamond round bur. The nasal floor was carefully elevated with the nasal mucosa attached without perforation. The size of the window was limited and kept small in order to not elevate the nasal floor too superiorly or medially. The dimensions of the window were controlled to primarily prevent post-operative volumetric changes of inferior meatus and to avoid any direct contact with the inferior turbinate. The nasal floor was elevated and did not exceed 10 mm into the medial aspect of the inferior meatus. Clinical visual inspection confirmed no perforation and an intact nasal mucosa. Simultaneous implant placement and biphasic calcium phosphate (Osteon III; Genoss, Suwon, Republic of Korea) was grafted into the site, as shown in Figure 2b. Implant fixtures included Implantium, 4.0 × 12 mm (Dentium, Suwon, Republic of Korea), at site #4 and the right upper maxillary second premolar with cover screw. Additionally, at site #2, the right upper maxillary second molar, the axial approach using Summers osteotomy was utilized and an Implantium, 4.0 × 10 mm (Dentium, Suwon, Republic of Korea), was placed with cover screw [8]. Buccal dehiscence defects, as noted in Figure 2c, were grafted with additional biphasic calcium phosphate bone graft. The grafted site was secured with a resorbable barrier collagen membrane measured at 30 × 20 mm (Genoss, Dentium, Suwon, Republic of Korea), as observed in Figure 2d. The reflected flaps were released buccally and palatally for tension-free closure, approximated, and sutured closed with 4-0 nylon (Ethilon^®^ 4.0, Ethicon, Cincinnati, OH, USA).

For post-operative management, medications were prescribed, including systemic antibiotics, Cefradine 500 mg twice a day for 14 days (Yuhan Pharmaceutical Co., Ltd., Seoul, Republic of Korea), and non-steroidal anti-inflammatory drug Anaprox 275 mg (Chong Kun Dang Pharmaceutical Co., Seoul, Republic of Korea), as needed. The patient was instructed to maintain oral hygiene with the use of 0.12% chlorhexidine solution (Hexamedine, Bukwang Pharmaceutical, Seoul, Republic of Korea) BID for 14 days. The patient was cautioned against expelling nasal mucus forcefully through her nose for 7 days and was also advised to keep the surgical area clean. Prior to her dismissal, a CBCT was acquired to obtain the post-operative view of the surgical procedure, including the location of the implants in relation to the inferior meatus (Figure 3a,b).

### 2.3. Radiographic Evaluation and Prosthetic Rehabilitation

Post-suture removal and after obtaining visual confirmation of soft tissue healing, a temporary removable prosthesis was provided to temporarily rehabilitate the patient for function. Six months after the initial surgery, the patient was scheduled for implant uncovering. Under local anesthesia, a mid-crestal incision with a single vertical incision mesial to the expected implant site was created. A buccal flap was exposed, as depicted in Figure 4b. For histological analysis, a 3 mm trephine drill was used to retrieve a core sample of augmented bone, approximately 6 mm in depth and distant from the implant sites, which was placed in a neutral 10% buffered formalin solution. The cover screws were replaced with healing abutments and the flaps were closed using 4-0 nylon (Ethilon^®^ 4.0, Ethicon, Cincinnati, OH, USA) and Chromic gut (Figure 4c). Post two months, after uncovering, a fixed eight-unit, one-piece, cement-retained prosthesis utilizing four fixtures was inserted, as depicted in Figure 4d. The histological sample retrieved was fixed, sectioned, and stained with Hematoxylin and Eosin (H&E), and visualized using a light microscope (BX-51, Olympus Optical, Tokyo, Japan). The stained slides were examined for the total bone volume area of the core, types of cells present, and tissue type.

## 3. Results

The case report presented here demonstrates no clear perforations with uneventful healing for the patient. Immediately following the surgery, CBCT analysis of the implant placement and bone augmentation demonstrated adequate elevation and grafting in the lateral nasal wall of the inferior meatus (Figure 3a,b). Figure 3a specifically demonstrates a well-defined periphery of the grafted site along the implant placed at site #4. The implant at site #2 was placed utilizing the transcrestal approach and depicts partial bony engagement of the lateral wall (Figure 3b). The surgical site healed uneventfully and no early or late complications were documented. Six months post-surgery, radiographic and clinical examination revealed stable implants engaged with the augmented bone during the uncovering. The bone biopsy harvested at implant uncovering and the placement of healing abutments were evaluated morphometrically. The core-stained slides were examined and depicted in entirety in Figure 5a, at 1× resolution. Figure 5b at 10× resolution, provides a detailed view of the newly formed bone and residual calcium phosphate particles. The newly depicted bone demonstrates osteocytes encased within and associated active osteoclast and osteoblasts cells. The actively remodeled sites are flanked by osteoid deposits and residue calcium phosphate particles. Histologically, the site does not demonstrate any necrotic bone, bacterial debris, or invasion. The site demonstrates active bone remodeling, indicating vital bone undergoing bone turnover where the biphasic calcium phosphate is eventually resorbed and replaced by new bone. Furthermore, clinical and radiographical observations reveal no associated complications such as marginal bone loss around the implants placed at sites #2 or #4 or soft tissue ingress, etc. (Figure 6a–c). The patient’s airways continue to remain patent, with no clinical complication or masticatory problems reported in subsequent follow-ups.

## 4. Discussion

Maxillary sinus augmentation via a lateral approach using the Caldwell–Luc osteotomy has been widely utilized to treat severely atrophic posterior maxilla for the placement of endosteal dental implants [9,10]. Similarly, anterior nasal floor elevation was demonstrated retrospectively with 32 patients and 100 implants to be a predictable procedure for significant atrophic sites [11,12]. Despite the minor differences in the clinical and cellular presentation of the nasal mucosa and sinus mucosa, with the former being thicker and more vascular than the latter, both the maxillary sinus and nasal cavities share the same pseudostratified ciliated columnar epithelium [13]. Thereby, the anatomic presentation combined with the presented case study demonstrates the nasal floor elevation as a viable augmentation procedure with the future potential to be utilized as a predictable approach. Placing implants in the posterior atrophic maxillary area when combined with the inferior meatus pneumatization represents a challenge for the clinician surgically. However, the anatomic variation when correctly visualized and diagnosed with a CBCT scan can be managed by employing nasal floor elevation. The current case study demonstrates an approachable technique, where the risk of perforation is low, and the augmentation provides an increase in bone height for the purpose of simultaneous implant placement. The approach of simultaneous maxillary sinus augmentation and implant placement has been shown to produce promising outcomes long-term. Cho et al. demonstrated successful bone formation around implants that were placed during sinus bone grafting with deproteinized bovine bone mineral [14]. In a human autopsy study, Lee et al. also showed long-term volume stability after sinus augmentation using biphasic calcium phosphate and simultaneous implant placement [15]. In the current report, the histological analysis demonstrates vital bone at the grafted site, indicating that the procedure yields new bone volume. The histological results extrapolated to the implant site combined with the clinical stability of the implants demonstrate implant osteointegration at a previous atrophy ridge.

Considering nasal airflow in a patient, the widening of the nasal cavity has been previously demonstrated to increase the airway space and improve nasal patency [16]. Conversely, when the distance between the lateral wall of the nasal cavity and the nasal septum is decreased, the resistance to nasal airflow increases, subsequently causing nasal respiratory problems in patients [17]. Thereby, when the lateral nasal floor elevation for inferior meatus pneumatization is performed as per this case study, the lateral breadth of the nasal cavity was predicted to become narrow, theoretically altering the airflow. However, the procedure did not cause any alteration in airflow and there were no documented airway issues or complications subsequent to the procedure for the patient.

To the best of the authors’ knowledge, the performed procedure—a nasal floor elevation with grafting to circumvent inferior meatus pneumatization—has not been reported. In the present case, the procedure was successfully performed and rehabilitation with dental implants and a final prosthesis was utilized to restore the patient’s masticatory function. Although the reported case was a single case and the follow-up period was only 2 years, no complications such as compromised airflow, breathing difficulties, or loss of implant were noted within this period. Clinical parameters such as pocket depths and bleeding on probing were also all within the normal limits. Histopathological findings of the present case also showed similar outcomes to those of sinus floor augmentation [18,19].

The authors recommend that clinicians evaluate patients routinely for inferior meatus pneumatization by employing a CBCT. Furthermore, on detection of the anatomic variant, pre- and post-operative co-evaluations with otolaryngologists may also help with an accurate diagnosis, treatment plans, and the management of any unforeseen complications from grafting inside the nasal cavity. The use of an endoscope may further help with a more detailed assessment of the surgical field itself. The authors hope that the presented case study piques further investigations on anatomic variants and how to circumvent atrophic ridges for the purpose of providing overall function for patients who need rehabilitation.

## Figures and Tables

**Figure 1 medicina-59-00357-f001:**
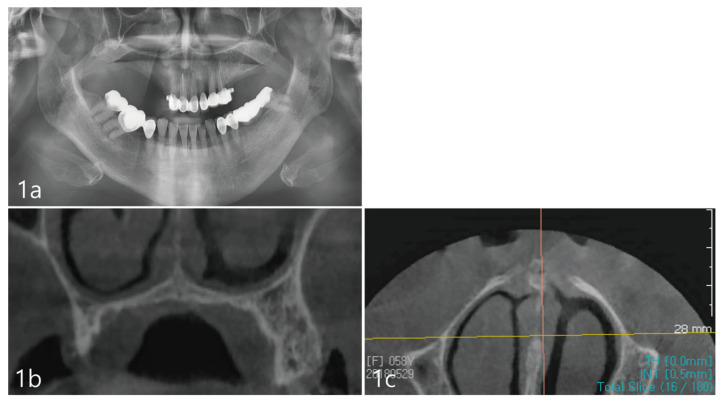
(**a**). Pre-operative panoramic radiograph shows severely atrophic posterior maxilla. The maxillary sinus is not well-observed. (**b**). Coronal view of the CBCT showing a portion of the maxillary sinus at the most upper-left corner. The ridge dimensions show deficient vertical and horizontal bone volume. (**c**). Axial view of the CBCT shows the inferior meatus of the nasal cavity.

**Figure 2 medicina-59-00357-f002:**
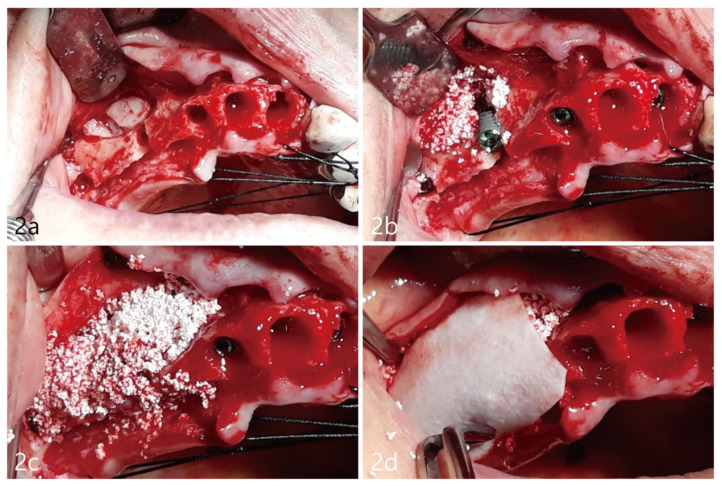
(**a**). A mucoperiosteal flap was reflected and an ovoid-shaped osteotomy was prepared on the buccal bone plate corresponding to the location of the inferior meatus. (**b**). Multiple fixtures were placed subcrestally, including an implant at site #4 implant. (**c**). Additional lateral bone augmentation was performed on both the buccal and palatal aspects. (**d**). An absorbable collagen membrane was placed at grafted sites to enclose the grafts.

**Figure 3 medicina-59-00357-f003:**
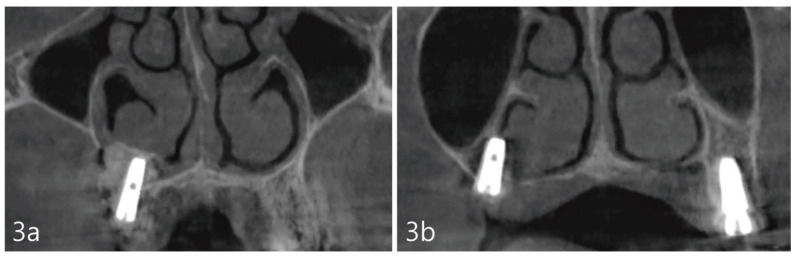
(**a**). Coronal views of the CBCT immediately after the implant placement showing adequate elevation and grafting with a well-defined periphery of the grafted site. (**b**). At site #2 tooth, the implant was placed with partial involvement of the lateral wall of the nasal cavity.

**Figure 4 medicina-59-00357-f004:**
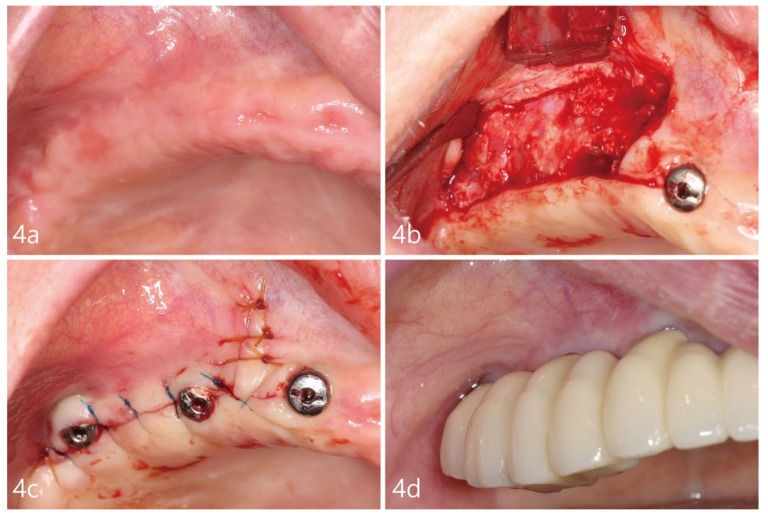
(**a**). Clinical photograph before uncovering of the fixtures. (**b**). New bone formation evident upon re-entry. A core was collected from an area away from the implants. (**c**). Placement of healing abutments and flap closures. (**d**). After 2 months of healing, a cementable final prosthesis was inserted.

**Figure 5 medicina-59-00357-f005:**
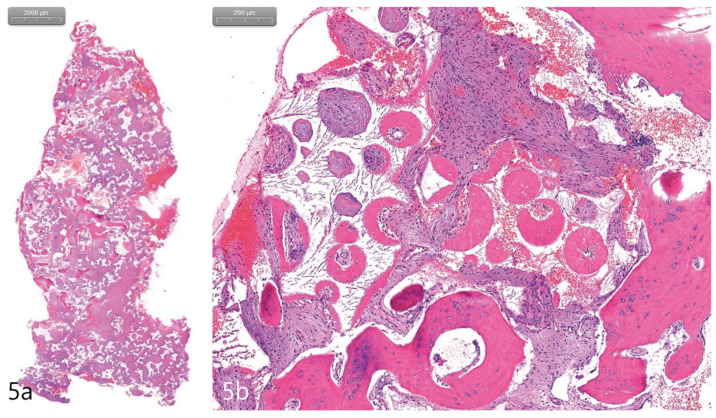
(**a**). The core biopsy with H&E stain. (**b**). The newly generated bone was well-observed around the bone substitute particle. The new bone contained osteoids, osteoblasts, and osteocytes. Immature osteoid tissue was distributed between bone grafts (H&E stain).

**Figure 6 medicina-59-00357-f006:**
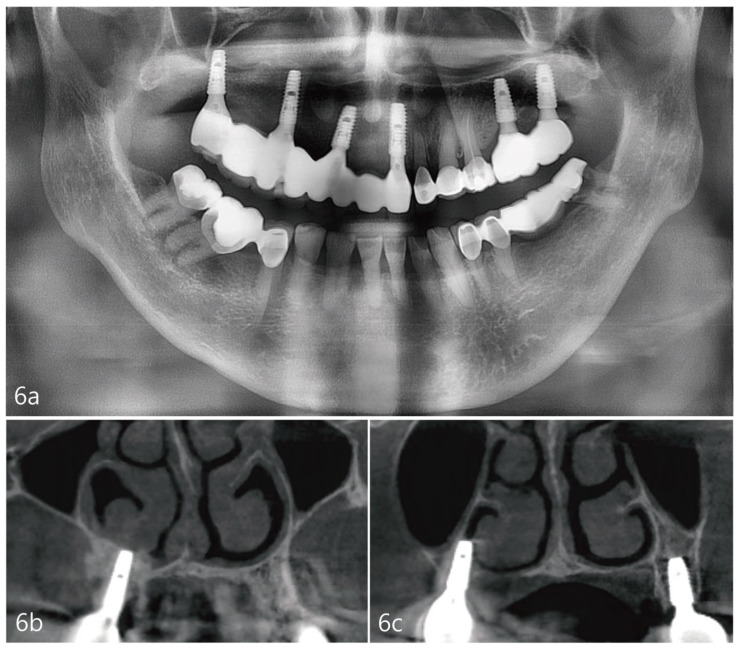
(**a**). Panoramic radiograph 2 years after prosthesis delivery. (**b**). Coronal view of the CBCT at the #4 site showing no marginal bone loss around the implant or other abnormal findings. (**c**). Coronal view of the CBCT at the #2 site showing no marginal bone loss around the implant or other abnormal findings.

## Data Availability

All data and material are presented in the manuscript.

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
