# Peer review of "Successful Implant Placement via Simultaneous Nasal Floor Augmentation in an Inferior Meatus Pneumatization Case"

_medicina, 2023, doi:10.3390/medicina59020357_

Round 1

Reviewer 1 Report

This case report is demonstrating the treatment of a maxillary partial edentulism via dental implants applied through sinüs and nasal lifting. The authors mention the nasal meatus and its augmentation. The authors retrieved a biopsy of the augmented bone and processed it histologically showing new bone formation and residues of the CAP material. This procedure may require an IRB board approval for being published (ethical approval) Nothing is mentioned in the manuscript accordingly.

The clinical photos are clear and high quality. However, this case report adds little to the knowledge of the augmentation procedures related to the nasal meatus. Apart from the paragraphs, there is no schematic or figural demonstration of this procedure which readers may benefit from.

Hence the final panoramic image shows some marginal bone loss around those implants that were iterated as healthy by the authors.

I suggest: 1-obtaining an IRB approvaÅŸl, 2- continuing these cases 3- reporting them as a case series by the histologic series which is going to be more scientific.

In its current form, it has little to no benefit for the readers.

minor remarks: Please correct the citation PARK WB in the introduction.

Author Response

Thank you very much for your comments and suggestions.  The authors’ responses are outlined below.   

I suggest: 1-obtaining an IRB approvaÅŸl, 2- continuing these cases 3- reporting them as a case series by the histologic series which is going to be more scientific.

In case reports and case series, formal ethical approval may not be appropriate or necessary. Nevertheless, the patient has signed the informed consent to participate in the project and to have the case published in an academic journal. All ethical issues have been considered to protect the patient’s rights.  A new paragraph was added to describe all these.

The authors also agree that a case series with more histologic analysis will be more valuable.  However, this was only one case and the authors believed that a case report was a good start.  

minor remarks: Please correct the citation PARK WB in the introduction.

The citation was deleted.

Reviewer 2 Report

The manuscript "Successful Implant Placement via Simultaneous Nasal Floor Augmentation in an Inferior Meatus Pneumatization Case" describes a successful surgical approach to overcome the obstacles to implant placement, which can be presented by a rare anatomic variation of the nasal cavity floor. As emphasised by the authors, the reader should always consider that such anatomic variants may be encountered during the procedure of implant planning and placement but can also be managed successfully.

To the authors, I kindly suggest a thorough re-read to correct some English language mistakes (some sytax and punctuation placement errors here and there in the Introduction and Results sections). I also suggest adding a brief explanation of the prothodontic rehabilitation treatment plan and why was it chosen (why were the teeth 12,11,21 extracted and why an implant retained prosthesis was in their opinion the optimal treatment approach). I noticed the misuse (or rather a confusion) of the terms success and survival throughout the text. Please note, that these two terms are defined differently and must always be differentiated. In the light of this notion, please consider improving the conclusions by adding some more critique on the results (is two-year follow up enough to consider the approach successful, also considering the study design - a single case). Also consider adding periimplant parameters (PPD, bleeding on probing etc.) to the results, since those really determine the absence of biologic complications and thus the success of treatment.

Author Response

Thank you very much for your comments and suggestions.  The authors’ responses are outlined below.   

 To the authors, I kindly suggest a thorough re-read to correct some English language mistakes (some sytax and punctuation placement errors here and there in the Introduction and Results sections).

The English language in the entire manuscript has been revised.

I also suggest adding a brief explanation of the prothodontic rehabilitation treatment plan and why was it chosen (why were the teeth 12,11,21 extracted and why an implant retained prosthesis was in their opinion the optimal treatment approach).

New sentences have been added and highlighted in the revised manuscript.

I noticed the misuse (or rather a confusion) of the terms success and survival throughout the text. Please note, that these two terms are defined differently and must always be differentiated.

The authors are aware of the difference between the two terms.  These terms were only used when citing specific studies, and authors tried to be consistent with the use of the same terminologies shown in their actual manuscripts.  

In the light of this notion, please consider improving the conclusions by adding some more critique on the results (is two-year follow up enough to consider the approach successful, also considering the study design - a single case). Also consider adding periimplant parameters (PPD, bleeding on probing etc.) to the results, since those really determine the absence of biologic complications and thus the success of treatment.

A paragraph was added to reflect the suggested comments.

Reviewer 3 Report

This is an interesting report that presented diagnosis and management of an unusual case, i.e., inferior nasal meatus pneumatization that was treated with nasal floor bone augmentation and implant therapy. The case is well-presented and well-reported. If postoperative medication included nasal decongestants, this should be detailed. Possible reasons for inferior meatus pneumatization should be discussed. Limitations of this case report should be indicated.  I suggest accepting this manuscript for publication.

Author Response

Thank you very much for your comments and suggestions.  The authors’ responses are outlined below.   

If postoperative medication included nasal decongestants, this should be detailed.

No nasal decongestants were used on this patient post-operatively.

Possible reasons for inferior meatus pneumatization should be discussed.

There is very limited available data on this topic and authors could not find any good reasons for this phenomenon.

Round 2

Reviewer 1 Report

The study performed actions which are not routine and conventional and solely experimental for this particular manuscript- without an IRB approval. The bone-biopsy and histologic process is not routine and conventional and requires an ethical approval for publication. Violation of this requirement may results with local and/or international penalties.

The changes have improved the manuscript but dur to aforementioned reasoons should not be published. 

Author Response

Dear Reviewer,

Thank you very much for your comments regarding the ethics committee approval.  As mentioned previously, for case reports and case series, when reporting on research that involves human subjects, human material, human tissues, or human data, authors must declare that the investigations were carried out following the rules of the Declaration of Helsinki of 1975 (https://www.wma.net/what-we-do/medical-ethics/declaration-of-helsinki/), revised in 2013. The patient was explained all the details of the process;  informed consent was obtained and a signed copy is also available if needed,  The authors' statement is also included in the manuscript. 

Round 3

Reviewer 1 Report

The revisions and declaration of procedures with relation to the IRB status is sufficient for publication.